# Impact of Nut Consumption on Cognition across the Lifespan

**DOI:** 10.3390/nu15041000

**Published:** 2023-02-16

**Authors:** Stephanie K. Nishi, Aleix Sala-Vila, Jordi Julvez, Joan Sabaté, Emilio Ros

**Affiliations:** 1Universitat Rovira i Virgili, Departament de Bioquímica i Biotecnologia, Unitat de Nutrició, 43201 Reus, Spain; 2Institut d’investigació Sanitària Pere Virgili (IISPV), 43201 Reus, Spain; 3Centro de Investigación Biomédica en Red Fisiopatología de la Obesidad y la Nutrición (CIBEROBN), Institute of Health Carlos III, 28029 Madrid, Spain; 4Toronto 3D (Diet, Digestive Tract and Disease) Knowledge Synthesis and Clinical Trials Unit, Toronto, ON M5C 2T2, Canada; 5Clinical Nutrition and Risk Factor Modification Centre, St. Michael’s Hospital, Unity Health Toronto, Toronto, ON M5C 2T2, Canada; 6Cardiovascular Risk and Nutrition, Hospital del Mar Medical Research Institute (IMIM), 08003 Barcelona, Spain; 7Clinical and Epidemiological Neuroscience Group (NeuroÈpia), Institut d’ Investigació Sanitària Pere Virgili (IISPV), 43007 Reus, Spain; 8Barcelona Institute for Global Health (ISGlobal), 08036 Barcelona, Spain; 9Center for Nutrition, Lifestyle and Disease Prevention, School of Public Health, Loma Linda University, Loma Linda, CA 92350, USA; 10Lipid Clinic, Endocrinology and Nutrition Service and Institut d’Investigacions Biomèdiques August Pi Sunyer, Hospital Clínic, 08036 Barcelona, Spain

**Keywords:** nuts, cognitive performance, brain health, dementia, lifespan, aging, epidemiological studies, clinical trials

## Abstract

Cognitive health is a life-long concern affected by modifiable risk factors, including lifestyle choices, such as dietary intake, with serious implications for quality of life, morbidity, and mortality worldwide. In addition, nuts are a nutrient-dense food that contain a number of potentially neuroprotective components, including monounsaturated and polyunsaturated fatty acids, fiber, B-vitamins, non-sodium minerals, and highly bioactive polyphenols. However, increased nut consumption relates to a lower cardiovascular risk and a lower burden of cardiovascular risk factors that are shared with neurodegenerative disorders, which is why nuts have been hypothesized to be beneficial for brain health. The present narrative review discusses up-to-date epidemiological, clinical trial, and mechanistic evidence of the effect of exposure to nuts on cognitive performance. While limited and inconclusive, available evidence suggests a possible role for nuts in the maintenance of cognitive health and prevention of cognitive decline in individuals across the lifespan, particularly in older adults and those at higher risk. Walnuts, as a rich source of the plant-based polyunsaturated omega-3 fatty acid alpha-linolenic acid, are the nut type most promising for cognitive health. Given the limited definitive evidence available to date, especially regarding cognitive health biomarkers and hard outcomes, future studies are needed to better elucidate the impact of nuts on the maintenance of cognitive health, as well as the prevention and management of cognitive decline and dementia, including Alzheimer disease.

## 1. Introduction

Cognitive health is a key component of healthy aging. Age-related cognitive decline and neurodegenerative disorders, such as dementia, which are a consequence of population aging and an increased lifespan, are a growing public health concern [1]. Dementia, with Alzheimer’s Disease (AD) as the most common type, is currently one of the top ten leading causes of mortality among all diseases and one of the major causes of disability and dependency among older people worldwide [1]. As cognitive decline and dementias have physical, psychological, social, and economic impacts, not only for the people directly affected, but also for their caregivers, families, and society at large, it is a key public health concern to address [1].

While neurodegenerative disorders tend to occur at older ages, brain development and cognitive health are impacted across the lifespan, from the fetus stage in pregnancy through adulthood [2]. Increasing evidence indicates that a high cognitive reserve, a healthy lifestyle, and the control of modifiable cardiovascular risk factors may reduce the risk of developing cognitive decline and dementia, including AD [3]. Of the lifestyle components influencing brain health, nutrition holds much potential. Nutrition may directly affect the brain or indirectly influence risk factors shared by cardiovascular and neurodegenerative diseases, thereby possibly having a substantial influence on cognition and the risk of dementia [4,5]. For instance, oxidative stress and inflammation are thought to play a major role in the initiation and progression of AD and other neurodegenerative disorders [6]. Antioxidant-rich foods and dietary patterns are potential strategies to counteract cognitive decline and AD and promote healthy aging. As proof, evidence is accumulating from both prospective studies and randomized controlled trials (RCTs) that adherence to plant-based dietary patterns rich in antioxidant foods, such as vegetables, fruits, whole grains, legumes, and nuts, is associated with a delay in age-related cognitive decline among older adults from diverse populations, as summarized in recent systematic reviews [7,8].

The Mediterranean diet (MedDiet) has been by far the most investigated dietary pattern for associations with brain health outcomes, with findings suggesting a protective association with cognitive decline [9,10,11]. Neuroimaging studies have further supported the association between increasing adherence to the MedDiet and greater brain volumes, lesser changes of brain atrophy, and the preservation of structural connectivity in healthy older adults. Within these investigations, higher intake of specific nutrients such as unsaturated fatty acids, antioxidant vitamins, and polyphenols has been linked to larger brain volumes [12]. More limited research on other healthy plant-based diets, such as the dietary approach to stop hypertension (DASH) diet, the Mediterranean-DASH diet, the intervention for neurodegenerative delay (MIND) diet, and other anti-inflammatory diets, has shown beneficial associations with cognitive health in older adults as well [13,14].

Given the available evidence, the World Health Organization (WHO) guidelines for risk reduction of cognitive decline and dementia included nutrition-related recommendations relating to the MedDiet and healthy, balanced, plant-based dietary patterns, all of which included nuts [15]. In addition, the WHO guidelines state that unsaturated fats, such as those found in nuts, are preferred over saturated fats for brain health and that consumption of nuts has been associated with a reduced risk of cognitive impairment [15].

Nuts (i.e., tree nuts, including almonds, Brazil nuts, cashews, hazelnuts, macadamias, pecans, pine nuts, pistachios, and walnuts; and peanuts) are an integral part of plant-based diets and have an optimal nutrient profile, being particularly abundant in anti-inflammatory and antioxidant molecules, such as unsaturated fatty acids, non-sodium minerals, vitamins, and polyphenols; moreover, their frequent consumption is associated with a consistent reduction in the risk of cardiovascular disease (CVD) [16]. Due to the fact that cardiovascular risk factors and CVD have well-established links to neurodegeneration and unhealthy aging, nut consumption, already well known to benefit vascular function, has been hypothesized to also favor cognition and overall brain health [17].

The present narrative review aims to present up-to-date evidence regarding the association between nut consumption and cognitive health during different stages of life. Specifically, it reports on the proceedings from the “Nuts 2022, Where we are and where we are going in research” international conference session titled “Nuts, Ageing, and Cognition.” In this session, epidemiological, clinical, and mechanistic evidence regarding nut consumption and cognition in different age groups was presented and discussed. This review is not a systematic review, and thus limitations should be acknowledged in that all studies may not have been identified. However, this review summarizes the available literature from independently conducted searches, and findings were further shared and discussed among the ensemble of experts in the field of nut and health research.

## 2. Nut Consumption and Neurodevelopment in Early Life (Gestation to Young Adulthood)

The early life years are critically important for cognitive development. Generally, brain development begins a few weeks after conception and is thought to be complete by early adulthood. The basic structure of the brain is believed to be shaped primarily during the prenatal period and early childhood, with the formation and refinement of neural networks and modification of functional abilities continuing over the long term [2]. Nutrition during each of the life stages of pregnancy, lactation, childhood, and adolescence can have a fundamental influence on development [18,19].

### 2.1. Nut Consumption and Prenatal Cognitive Development

The period of in-utero growth during gestation is considered to be particularly important for neurodevelopment since the brain undergoes several uniquely intense and complex processes [20,21]. Human brain development begins soon after conception with the inception of the formation of the neural tube and continues into early adulthood. The fetal brain begins to develop during the third week of gestation. By the end of the embryonic period (gestational week 10), the basics of the neural system are established. All the structures continue to develop throughout the fetal period and early childhood. By 6 years of age, the brain has reached 90% of its adult volume [22]. During this period of brain development, essential nutrients, such as the omega-3 fatty acid alpha-linolenic acid (ALA), which can be found in nuts, particularly walnuts, may alter the epigenetic control of neural processes, neuron formation, migration, axon and dendritic growth, synaptogenesis, and myelination [23]. In the long term, adequate nutrition that promotes neurodevelopment during the in-utero period may benefit a child’s neuropsychological development, school performance, and future professional success [21]. Yet, the possible protective effects of nut consumption on cognitive health have hardly been explored in child neurodevelopment. At present, there is limited evidence from epidemiological and clinical studies assessing nut consumption and brain function during pregnancy, childhood, or adolescence.

In relation to the pregnancy period, to our knowledge, only one study has been published assessing the association between cognitive health and nut consumption during early development. This study involved the Spanish Childhood and Environment (Infancia y Medio Ambiente, INMA) Project, a large prospective, multicenter, population-based cohort of 2208 mother-child pairs, conducted in several regions of Spain [24]. Mothers were followed during pregnancy (first and third trimesters), and their children were enrolled at birth and followed until the age of 8 years. Twice during pregnancy and at the children’s ages of 1.5, 5, and 8 years, dietary intake questionnaires and neuropsychological assessments were administered. The mean nut consumption among mothers in the first trimester of pregnancy was 41 g/week (standard deviation [SD], 74 g/week), and the median was 17 g/week (interquartile range [IQR]: 0 to 46 g/week), with a third of the total participants being non-consumers (*n* = 860, 33.5%). Overall, the authors found that higher maternal consumption of nuts in early pregnancy was associated with enhanced cognitive development in their children, compared to non-consumers, at 1.5, 5, and 8 years of age [24].

### 2.2. Nut Consumption and Early Life (Childhood and Adolescence) Cognitive Development

Nutrition during childhood is particularly important, as this is a period of relatively rapid brain development, and nutrients aid the brain in the creation of new synaptic connections during learning processes at school and in home environments [18]. While the structural components of the brain and the foundations of basic sensation and perception systems are fully developed by the time children reach kindergarten age, other systems such as those involved in memory, decision-making, and emotion continue to develop well into childhood. The foundations of many of these abilities, however, are constructed during the early years. Whereas the functional aspects of the brain can have varying developmental time frames and patterns, adolescence is an important period of brain development and remodeling to functionally develop for thinking and processing [2]. The brain reorganizes during this developmental stage with functional and structural changes resulting from the re-emergence of gonadotropin-releasing hormone, triggering a cascade of hormone-dependent processes. Other biological processes involve epigenetic factors, which are highly sensitive to the environment and may therefore make this period of growth more vulnerable to external insults [25]. Moreover, the prefrontal cortex, which carries out important functions such as internally guided behaviors (control of emotion), logical thinking, working memory, and organizing skills (executive function), is the last region of the brain to mature (in the early twenties). The synaptic plasticity of the prefrontal cortex is accentuated during adolescence, a process that involves loss of grey matter density and an increase in white matter volume, cerebral blood flow, and synaptic pruning. Adolescence is also a time of refinement of brain connectivity and complex behaviors [26]. It is widely recognized that the synaptic plasticity of the brain decreases with age [25,26], but this pattern does not seem to follow a linear trend, and adolescence is an important period during which brain development can be enhanced and protected from environmental hazards, from air pollution to unhealthy diets, with long-term consequences.

Considering the importance of brain structural and functional development during childhood and adolescence, very few studies have assessed the association between nut consumption and cognitive health during these life stages. One cross-sectional study conducted in 317 Korean children and adolescents (167 girls and 150 boys) with a mean age of 11.8 (range, 6 to 18) years and no prior diagnosis of neurologic or psychiatric disorders assessed the consumption of nuts, among other healthy foods estimated from diet questionnaires, in relation to cognitive performance [27]. Nut consumption was related to improved cognitive reaction time consistency and attention function as measured by the symbol-digit modality test (SDMT). However, no associations were observed with the other neuropsychological measures, specifically the verbal and visual memory tests, the shift attention test, the reasoning test, and the digit span forward and backward tasks, assessed as part of a computerized cognitive assessment battery. A limitation of this study is that the authors did not adjust the data for covariables known to influence cognitive performance in youth, such as parental social class, parity, type of delivery, breastfeeding, birthweight, maternal intellectual quotient (IQ), maternal mental health, maternal smoking and alcohol intake during pregnancy, clinical history during pregnancy, overall dietary pattern, and stress events [27].

While there is currently a lack of RCTs, there is promise for further evidence as a protocol for a RCT (the WALNUT study [WSS]) aiming to assess the effect of daily walnut consumption (30 g) over 6 months on cognitive function among nearly 700 healthy adolescents from several high schools in Barcelona, Spain, was recently published [28]. The results of this study will eventually fill a gap in our knowledge of the effect of nut consumption on cognitive health in adolescence.

Linking the previous findings relating the consequences of increased nut consumption by mothers during pregnancy to the later neuropsychological traits of their children [24], and further applying public health recommendations to an entire population, for example, recommending pregnant women to consume a daily serving of nuts throughout the prenatal period, might be hypothesized to increase the population mean IQ score by a few points. This is not clinically significant at the individual level; still, the impact on the IQ distribution at the population level might represent a significant reduction in the proportion of children with learning problems or low IQ scores [24,29]. Depending on the findings of future research, this could be a potentially impactful health promoting message for the population, possibly akin to the promotion of long-term breastfeeding and/or the recommendation to stop smoking during pregnancy.

### 2.3. Nut Consumption and Cognitive Health in Young Adulthood

As adolescence transitions into young adulthood, complex cognitive behaviors are refined. It is during this life stage that myelination of regions involved in higher cognitive abilities, such as the prefrontal cortex, is considered to be complete [2]. This process of myelination ultimately involves the axons of neurons being wrapped in fatty cells, which facilitates neuronal activity and communication for the transmission of electrical signals.

Nutrition, and particularly fatty acids, are critical for brain development. Essential fatty acids are long-chain polyunsaturated acids (LC-PUFAs) that the body cannot synthesize and must be obtained from the diet (mainly from oily fish, seeds, and nuts). LC-PUFAs are involved in the function and architecture of the central nervous system throughout the various life stages. It has been shown that the LC-PUFA docosahexaenoic acid (DHA) regulates neurotransmission systems such as serotonergic, dopaminergic, norepinephrinergic, and acetylcholinergic systems [23,30].

The omega-3 PUFA ALA, which is particularly abundant in walnuts, has been associated with cognitive function in older people [31]. However, it has been scarcely studied at younger ages. In a recent cross-sectional study of 332 healthy adolescents, the red blood cell proportions of ALA (an objective biomarker of walnut consumption) were inversely associated with impulsivity (a usually detrimental psychological trait and a key feature of many psychiatric disorders) [32]. The findings from this study support the hypothesis that nuts, particularly walnuts, could have a beneficial impact on cognition.

Additionally, to date, the only known RCT specifically investigating the effect of nut consumption on cognitive health in youth was that conducted by Pribis and colleagues in young adults aged 18 to 25 years [33]. In a crossover investigation, the consumption of 60 g of walnuts for 8 weeks by college students (*n* = 47) was associated with better critical thinking abilities as measured by the Watson-Glaser Critical Thinking Appraisal. However, no differences were observed for verbal reasoning, as measured by Raven’s advanced progressive matrices, or memory, as assessed by the Wechsler Memory Scale, when compared with the placebo group. The authors acknowledged that these findings may be limited by the short duration of the intervention as well as by the fact that participants were college students, whose baseline cognitive functioning may be higher than that of the general population and hence may impact cognitive findings.

Early-life structural and functional development of the brain and the potential influence of dietary intake are important, yet little research has been conducted to assess the impact of nut consumption on cognitive health during childhood and adolescence. Although preliminary research is promising, further investigations are warranted to better determine the efficacy of consuming nuts on neurodevelopment, especially during life stages of growth and development.

## 3. Nut Consumption and Cognitive Performance in Adulthood

An essential component of healthy aging is normal cognitive function, which critically affects functional independence and health-related quality of life. Increased life expectancy and subsequent population aging entail a rising prevalence of age-associated cognitive impairment, a major public health concern given its frequent transition to mild cognitive impairment (MCI) and dementia, including AD [34]. Indeed, many older adults experience deteriorating cognitive function, usually with declining episodic memory and executive function that parallel volume losses in critical brain structures such as the hippocampus [35]. Cognitive domains that can be interrogated with specific neuropsychological tests include memory, executive function, attention, language, and visuospatial skills. The neuropsychological test most commonly used in epidemiologic studies is the Mini-Mental State Examination (MMSE), a brief test (it takes 7 to 10 min to complete) that is very useful to detect dementia when it is grossly abnormal but is limited in its ability to provide insight into subtler and much more frequent cognitive deficits [36]. A telephone-adaption of the MMSE (the TICS, or Telephone Interview for Cognitive Status) to assess overall cognitive performance has also been frequently used in the absence of more comprehensive tests. While these screening instruments may offer an opportunity for cognitive comparisons, their accuracy in detecting cognitive impairment is a limitation in neurocognitive studies. MCI is said to be present when there is objective evidence at cognitive testing that one or more of these cognitive domains is impaired. As opposed to individuals with dementia, those with MCI maintain their independence in functional abilities and have no significant impairment in social or occupational functioning [37].

### 3.1. Epidemiological Studies Examining the Association of Nut Consumption with Cognitive Performance

In the review of epidemiological evidence evaluating nut consumption and cognitive performance, 15 studies were identified, including 7 cross-sectional and 8 prospective cohort studies. Table 1 lists these investigations by date of publication and summarizes their findings. Briefly, these observational studies involved men and women, with the majority aged ≥50 years, from Australia, China, Italy, the Netherlands, Norway, Spain, and the United States, with the prospective cohorts ranging from 3 to 20 years in duration and assessing quantiles of nut intake comparing none or low to higher intake dosages. The known factors influencing cognitive performance in adulthood, namely age, sex, educational level, body mass index, cardiovascular risk factors (smoking, hypertension, dyslipidemia), physical activity, overall dietary pattern, and depression [3], were treated as confounders and adjusted for in analyses of data from epidemiological studies.

Out of a total of 15 observational studies, 13 showed a positive association between nut consumption and cognitive performance; however, beneficial relationships were not observed among all cognitive assessments conducted in each study. For instance, in a prospective Dutch study of middle-aged adults, cognitive performance was assessed at baseline and after a 5-year follow-up in relation to quintiles of consumption of plant foods. The highest nut intake was associated with better cognitive function (i.e., memory, speed, flexibility, and global cognitive function) at baseline but not lesser cognitive decline at follow-up when data were adjusted for cardiovascular risk factors [39]. In a cross-sectional study of Chinese adults, higher consumption of fruit, vegetables, and nuts was associated with delayed memory, while all other cognitive domains were unaffected; moreover, cognitively healthy participants consumed more nuts than those with MCI [44].

Further cross-sectional analyses have presented significant beneficial relationships between nut consumption and cognitive health. A cross-sectional study of nut consumption and cognitive performance was nested within a sub-cohort of the Prevención con Dieta Mediterránea (PREDIMED) study. This was a landmark 5-year RCT in which a MedDiet supplemented with either extra-virgin olive oil or mixed nuts (30 g/day: 15 g walnuts, 7.5 g almonds, and 7.5 g hazelnuts) versus a control diet (advice to follow a low-fat diet) resulted in nearly a 30% reduction in CVD events in older individuals at high cardiovascular risk [53]. The cross-sectional sub-study assessed the association of consumption of various foods with cognitive function. Of all the foods considered, only olive oil, coffee, wine, and walnuts—but not total nuts—were related to better cognitive function independently of known risk factors for cognitive decline, other food consumption, and energy intake. Of note, total urinary polyphenol excretion, an objective biomarker of intake of polyphenol-rich foods, was directly associated with working memory function [40]. In addition, within the context of the MedDiet, consumption of 1 serving of nuts (30 g)/week by older Italian adults was cross-sectionally associated with a reduced risk of low MMSE [45]. Cross-sectional analyses of the U.S. National Health and Nutrition Examination Survey (NHANES) indicated beneficial associations between nut consumption and cognition, determined based on 24-h dietary recalls [43,48]. Findings from the most recent relevant NHANES analyses of participants aged 65 and older who consumed nuts 15 to 30 g/d or met recommendations by consuming >30 g/day (either group accounting for 10% of the cohort) had better cognitive scores than non-consumers or low consumers [48]. A moderately sized cross-sectional study within the Sydney Memory and Ageing Study related consumption of different food groups to cognitive performance and found that higher consumption of nuts and legumes together related to higher global cognition, visuospatial function, and language [51].

Furthermore, prospective cohort analyses have demonstrated a potential beneficial relationship between nut consumption and cognitive function. In brief, a U.S. prospective study of a large sub-cohort of older women from the Nurses’ Health Study (NHS) assessed total nut consumption in relation to cognitive function and found an association with better average status for all cognitive outcomes analyzed. The difference in the global composite score between women consuming at least 5 servings of nuts/week and non-consumers was equivalent to the mean difference observed in cognitive status between women 2 years apart in age [42]. Another sizable U.S. cohort framed within an observational study of an aging population found that participants with any walnut consumption had greater cognitive scores at baseline, but no association with 4-year cognitive change was observed. However, only 13% of the sample had moderate walnut consumption of around ½ oz (14 g) per day [50]. Along similar lines, three large Asian prospective cohort studies showed favorable associations between nut intake and cognitive function. A prospective large 15-year Chinese study of participants in a nutrition survey aged 55 years or more found that consumers of nuts (mostly peanuts), which made up only 17% of the cohort, had less cognitive decline, as measured with the telephone version of the MMSE, than those not consuming any nuts [46]. The large prospective Singapore Chinese Health Study, in which nut consumption was determined at baseline and cognitive function was measured by the MMSE after 20 years of follow-up, reported an inverse association between graded nut consumption and reduction of cognitive function [49]. However, adjustment for intake of unsaturated fatty acids attenuated the association to non-significance, suggesting mediation of the cognitive effect by this key nut component. Finally, a large Chinese cohort study assessed nut consumption at baseline and in relation to changes in the MMSE, administered one to three times during a 6-year follow-up; the results showed that higher nut consumption related to a lower risk of cognitive impairment [52]. Conversely, possibly due to the shorter duration, a small prospective Italian study reported that baseline nut consumers versus non-consumers had a borderline reduced rate of developing a low MMSE after follow-up for 3 years [47].

Regarding specific types of nuts in general, walnuts appeared to be the type most studied and reported to be associated with better cognitive function when compared to low or non-consumers [41,43,47]. However, most of the studies assessed the total nut consumption as a whole and did not or could not delineate the analyses by the different types of nuts.

Two of the 15 studies did not show statistically significant associations between nut consumption and cognitive performance. Specifically, in a cross-sectional investigation of an older Norwegian cohort, nuts were non-significantly associated with better cognitive performance, although only 16% of the participants were nut consumers [38]. Additionally, a large U.S. prospective cohort, the Women’s Health Study, found no association for changes in cognitive performance in relation to quintiles of total nut consumption over a 4-year follow-up [41].

Overall, the epidemiological evidence suggests nut consumption may be positively associated with cognitive health. Still, the quality of the evidence from these epidemiological studies, which cannot determine causation, may be considered low for several reasons. First, six of the 13 studies with positive results were cross-sectional, and in four investigations reporting a beneficial association between increased exposure to nuts and cognitive performance, the outcome was assessed with the MMSE, exclusive of more precise neuropsychological tests. Second, most studies obtained exposure data from food frequency questionnaires (FFQs) or diet recalls, which have inherent weaknesses with regard to possible measurement error and recall bias [54]. Third, most prospective studies have an additional problem, i.e., nut exposure is assessed only once at baseline, thus missing the effect of any changes in consumption during follow-up. Fourth, tree nut and peanut data were also often reported in combination in the assessments, and information on nut preparation (salting, roasting, etc.) was lacking. Fifth, based on the available data, the prevalence of nut consumers was usually low, with some cohorts reporting only 13 to 30% of the study population consuming at least 1 serving (28 g)/week. Only 2 studies evaluated cohorts that met nut consumption recommendations of 30 or more g/d compared to non-consumers [44,45]. Finally, the amounts of nuts consumed by consumers tended to be relatively low (e.g., 2.9 g/d to ≥20 g/d) or were not sufficiently described to be able to provide comprehensive assessments and interpretations to help inform practice.

### 3.2. Randomized Controlled Trials of Nuts with Outcomes on Cognitive Performance

The results of scientifically sound RCTs are critical for formulating evidence-based dietary recommendations. However, few RCTs have examined the effects of nuts on cognitive outcomes in adults, and even fewer had sufficient statistical power or an intervention period lasting more than 6 months (Table 2). Hence, the level of evidence is still fragmentary.

Additionally, two of the largest RCTs with the longest follow-up were sub-studies of the PREDIMED trial. In a study conducted in the Navarra recruiting center, two neuropsychological tests assessing general cognition were administered [55]. However, these tests were only administered at the end of the study, following a median 6.5-year intervention period, thus changes over time were not evaluated. The results indicated that the two MedDiets (enriched with olive oil or mixed nuts) were associated with better cognitive outcomes compared to the control diet. In another PREDIMED sub-study carried out in the Barcelona center, a comprehensive cognitive battery was administered both at baseline and at the end of the trial after a median follow-up of 4.1 years [56]. The results showed that values for all cognitive domains declined in participants randomized to the control diet, while composites of memory performance, executive function, and global cognition improved above baseline with the two MedDiets. However, the improvement in executive function and global cognition observed with the nut diet did not reach statistical significance compared to the control diet. The findings demonstrated that a MedDiet supplemented with mixed nuts could delay the age-related decline of memory function.

Two small, short-term RCTs investigated the effect of peanuts [57] and almonds [58] on outcomes of cognitive performance in individuals with overweight or obesity. Surprisingly, given the intervention only lasted 12 weeks, in the study of Barbour et al. [57], the diet enriched with high-oleic acid peanuts resulted in improvements in short-term memory and verbal fluency compared to the control diet. In this trial, blood flow velocity in the middle cerebral artery was measured non-invasively with transcranial Doppler, and results showed that the peanut diet increased cerebrovascular reactivity (i.e., improved endothelial function of brain arteries). On the other hand, in the study by Dhillon et al. [58], almond consumption had no effect on cognitive performance compared to the control diet. In this trial, acute experiments examined whether a high-fat lunch rich in almonds would influence the well-known post-lunch dip in alertness, memory, and vigilance. The findings revealed that, compared with a high-carbohydrate meal, almond consumption at lunch counteracted in part the postprandial decline in memory, but not that of attention performance [58].

The large walnuts and healthy aging (WAHA) trial tested the 2-year effects of walnut consumption at 15% of daily energy on cognitive performance in healthy older adults from two sites, Barcelona (Spain) and Loma Linda (CA, USA) [59]. The WAHA study failed to find any differences in neurocognitive test scores for perception, language, memory, and frontal function domains or in a composite score of global cognition compared to the control diet. However, post hoc analyses by site revealed improved cognition in participants allocated to the walnut diet in Barcelona, who were more at risk of cognitive impairment than their California counterparts due to lower educational levels and more smoking. Functional brain magnetic resonance imaging (MRI) in a subset of the Barcelona cohort supported the benefit of walnuts on cognition [59]. Finally, a recent small RCT using different doses of almonds for 6 months in cognitively healthy middle-aged and older adults assessed with a complete neuropsychological test battery at baseline, 3 months, and 6 months found no differences in cognitive measures over time [60].

In summary, the findings of the two well powered, long-term, PREDIMED sub-studies examining the effects of MedDiets supplemented with nuts on cognitive performance indicated a beneficial effect in older individuals at high risk of CVD (thus, also at high risk of cognitive impairment and dementia); however, improved cognitive health might not be entirely attributable to nuts, as other components of the MedDiet changed in these studies [55,56]. Nevertheless, the MedDiet enriched with nuts reduced the relative risk of stroke by nearly 50% in the PREDIMED trial [39], which further supports the neuroprotective effect of nuts. Indeed, preventing stroke, post-stroke cognitive impairment, and dementia is critical for achieving optimal brain health [61]. The large, 2-year WAHA trial uncovered a salutary cognitive effect of walnuts only in participants at higher risk of cognitive impairment [59]. These findings concur with data collected in large multi-domain trials suggesting that individuals at high risk of cognitive impairment or who already have memory complaints, or MCI, are those who should be targeted for preventive interventions because they might obtain the largest benefit [62]. Clearly, larger and longer-term studies with nuts in populations at risk of dementia are warranted.

## 4. Potential Mechanisms of Action of Nuts in Cognitive Health

Normal aging involves many structural and functional brain changes. There are several hallmarks of cerebral atrophy (ventricular enlargement, cortical thinning, sulcal widening, and volume loss [63]), which are observed in parallel with declines in processing speed and certain memory, language, visuospatial, and executive function abilities [64]. Ageing is the primary risk factor for late-onset Alzheimer’s disease (AD; occurring in people aged 65 and over) [65]. However, AD should not be considered part of normal aging [66]. What determines the transition from normal aging to AD remains to be elucidated, but there is a long-standing consensus supporting the view of AD as a multifactorial disease [67], with pathological brain changes taking place years (even decades) before symptomatology is present. However, knowledge of AD is rapidly evolving. For many years, AD was conceived as a clinical-pathological construct, defined by the presence of symptoms/signs. The advent of cost-effective biomarkers prompted the re-definition of AD as a clinical-biomarker construct, referring to an aggregate of neuropathologic changes that can be identified in vivo much before clinical symptoms appear [68]. This preclinical phase of AD represents a therapeutic window for preventive strategies, which are of utmost importance, as highlighted by several international organizations [69,70].

Evidence is accumulating on the many diet components that might have a significant impact on the progression and prevention of AD (reviewed in [71]). As discussed, nuts are rich in compounds with anti-inflammatory, antioxidant, and hypolipidemic effects, thereby reducing the risk of CVD [16]. Given that CVD and AD share many risk factors, particularly hypertension, obesity, type 2-diabetes, and smoking [3], it is reasonable to assume that regular nut consumption might also protect against AD. Abundant experimental research supports this hypothesis (reviewed in [72]). As summarized in the preceding section, epidemiologic evidence, albeit generally of low quality, also concurs with this view, while RCTs of nuts for cognitive outcomes are incipient and confined to changes in cognitive performance after short- or medium-term supplementations. In this section, we will summarize possible mechanisms underlying the putative effects of nut bioactives on the two major hallmarks of AD, namely the buildup in the brain of amyloid-beta (Aβ) plaques and neurofibrillary tangles. We will review experimental research involving either nuts, nut extracts, or nut bioactives given on their own. Additionally, and concurrently with the current needs for research on diet and dementia [73], we will also focus on how available biomarkers might help in better selecting participants to be included in RCTs, and/or as intermediate endpoints, allowing for the detection of subtle changes after short-term interventions.

### 4.1. Nuts and Extracellular Plaque Deposits of Aβ

#### 4.1.1. Pathophysiologic Overview

According to the amyloid hypothesis [74], derangements in the production, accumulation, or disposal of Aβ are the main cause of AD. Aβ is a ~4 kDa peptide derived from the so-called β-amyloid precursor protein (APP), which is a transmembrane molecule. The enzymatic processes involving the metabolism of APP to Aβ involve a sequential cleavage by two membrane-bound endoproteases, β- and γ-secretase. In the first step, believed to be the rate limiting one, β-secretase (also known as BACE1) cleaves APP to release a large, secreted derivative (soluble peptide APPβ), while a 99-amino acid fragment (C99, also known as the carboxy-terminal fragment of beta [CTFβ]) remains bound to the fatty acid membrane. C99 undergoes a second cleavage by the action of γ-secretase, generating different species of Aβ, those ending at positions 40 (Aβ40) and 42 (Aβ42) being the most abundant ones (~80–90%, and ~5–10%, respectively) (reviewed in [75]). Aβ42 is more hydrophobic than Aβ40 and rapidly aggregates to form monomers and then mature fibrils and dense fibril meshes (senile plaques), the best-known hallmark of AD. However, soluble dimers, trimers, and small oligomeric Aβ42 aggregates other than monomers are increasingly believed to be more neurotoxic than Aβ42 mature fibrils [76]. Aβ deposition spreads from temporobasal and frontomedial areas to the remaining associative neocortex, primary sensory-motor areas, and the medial temporal lobe [77,78].

#### 4.1.2. Nuts and Aβ

Nut bioactives could hamper Aβ plaque build-up by targeting BACE1, γ-secretase, and/or Aβ aggregation (Figure 1A). To the best of our knowledge, while there has been exploration of the association between adherence to the MedDiet and AD biomarkers [79], there has been no assessment of nut consumption and AD, and there are no available clinical nut supplementation studies on this specific topic. In a cellular model of early AD (human SH-SY5Y cells transfected with APP695), treatment with 10 µg/mL of a lipophilic walnut kernel extract for 24 h resulted in a significant reduction in Aβ40 levels when compared to control cells [80]. In line with this finding, in a study testing the in vitro inhibitory effects against BACE1 of 18 different hydroalcoholic ethnomedicinal plant extracts (1 g per 5 mL), leaves (nuts were not tested here) from *Juglans regia* (walnut tree) were found to display a BACE1 inhibitory activity in a concentration-dependent manner [81]. In another in vitro study, the methanolic extract of walnut kernels (4 g per 10 mL) was found to inhibit Aβ fibril formation and defibrillate preformed Aβ fibrils [82]. Finally, reduced Aβ burden was described in experimental studies testing isolated bioactives found in nuts, including oleic acid [83,84], linoleic acid [84], ALA [85], beta-sitosterol [86], nicotinamide [87], ellagic acid [88,89,90], epigallocatechin [91], myricetin [92,93], caffeic acid [94], and an array of other polyphenols [93].

#### 4.1.3. Potential of Aβ Biomarkers in Future Research on the Neuroprotective Properties of Nuts

The accumulation of Aβ is considered the first detectable change of AD in the brain. It follows that an accurate determination of Aβ is relevant for the etiological diagnosis and for monitoring disease progression. Importantly, it might be a useful tool to identify changes in the rates of amyloid deposition without requiring long-term dietary interventions. Cerebrospinal fluid (CSF) analyses can indicate the presence of amyloid pathology in its earliest stages, although there is growing pressure to develop and validate less invasive blood-based biomarkers. A characteristic feature of AD is the reduction in CSF Aβ42 [95], which becomes evident about 15 years before clinical symptoms appear [96], although it plateaus relatively early in the AD continuum. The ratio Aβ42/Aβ40 in CSF has been repeatedly proven to improve prediction of clinical progression better than Aβ42 alone [97]. On the other hand, positron emission tomography (PET) imaging enables the non-invasive, in vivo assessment and quantification of continued build-up of amyloid burden beyond the CSF plateau, as well as providing information on the spatial distribution of the pathology in the brain [98]. Despite the potential interest of the Aβ42/Aβ40 ratio in CSF (or blood) and amyloid-PET imaging, none of them have been used as secondary outcomes in RCTs of nut supplementation with cognitive changes as primary outcomes.

### 4.2. Nuts and Neurofibrillary Tangles

#### 4.2.1. Pathophysiologic Overview

The second histopathologic hallmark of AD is the presence of neurofibrillary tangles, which are entirely made up of hyperphosphorylated tau protein. Tau is a microtubule-associated protein that promotes the formation of axonal microtubules, stabilizing them [99]. Tau might undergo phosphorylation at over 70 potential sites [100]. In AD, there is an abnormal tau hyperphosphorylation, which decreases the capacity of the protein to bind microtubules, promoting the destabilization of axons. Hyperphosphorylation also contributes to the detachment of tau, which self-aggregates to form paired helical and straight filaments, leading to the formation of intracellular neurofibrillary tangles [101]. These tangles are initially found in the entorhinal region and subsequently progress to the limbic system and neocortical regions, greatly correlating with cognitive decline [102].

#### 4.2.2. Nuts and Neurofibrillary Tangles

There is a current need to identify interventions capable of reducing tau aggregation by means of stabilizing microtubules, inhibiting tau phosphorylation, or inhibiting fibrilization. As in the case of Aβ, there is no clinical research on nut consumption and tau protein, while experimental research has tested only the bioactives that are present in nuts in isolation. Figure 1B presents the potential impact of specified nut bioactives on tau hyperphosphorylation. One of the most investigated is ALA, the vegetable omega-3 PUFA abundant in walnuts [16,31], which was found to inhibit tau aggregation in an in vitro study [103]. Furthermore, N9 (microglia) cells exposed to ALA increased their ability to better target [104], phagocyte, and degrade extracellular tau [105]. Another bioactive present in nuts tested in relation to tau is caffeic acid. In a study conducted in high-fat diet-induced hyperinsulinemic rats, the administration of caffeic acid (30 mg/kg body weight/day) for 30 weeks significantly reduced the expression of phosphorylated-tau protein in the hippocampus [94]. Similarly, pretreatment of P12 cells with caffeic acid prior to challenge with Aβ attenuated tau phosphorylation [106]. Finally, in a study including both in vitro (primary culture of cortical neurons) and in vitro (APP/PS1 double transgenic mice of AD), exposure to vitamin E reduced the formation of hyperphosphorylated tau through the inhibition of p38MAPK [107].

#### 4.2.3. How Biomarkers Can Help in Future Research

Akin to the use of biomarkers to quantify amyloid burden, the tau landscape is rapidly evolving, with ultrasensitive immunoassays allowing the reliable measurement of tau biomarkers in blood, while second-generation tau-PET tracers are being developed [108]. Patients with AD show increased levels of total-tau and phosphorylated-tau in CSF and blood when compared to healthy controls. Phosphorylated-tau is a more specific AD biomarker than total-tau, which can be increased in neurodegenerative diseases other than AD. Phosphorylation at threonine 181 (so-called p-tau181) is the most widely used tau biomarker, although other tau species, including those phosphorylated in other mid-domain residues (threonine 217, threonine 231), are increasingly used [95]. However, to the best of our knowledge, no RCTs have assessed changes in these biomarkers using nut supplementation.

## 5. Summary and Future Directions

Epidemiological, clinical, and mechanistic evidence, while limited and inconclusive, suggests a possible role for nuts in the maintenance of cognitive health and prevention of cognitive decline in individuals across the lifespan, particularly in older adulthood. Given the potential beneficial impact of nuts on cognitive health, their consumption within a healthy dietary pattern may offer a simple public health strategy for the prevention of cognitive decline in most individuals.

Still, the limitations of the presently available evidence should be acknowledged and considered for future research. There is a dearth of research on nuts and cognition, especially in individuals under 60 years of age, and no study has examined whether nut containing diets influence hard clinical outcomes (e.g., dementia or AD) [109]. On the basis of the strength of the evidence on nut/walnut consumption and heart disease, the US Food and Drug Administration issued qualified health claims for nuts in 2003 [110] and for walnuts in 2004 [111], stating that “supportive but not conclusive research shows that eating 1.5 ounces per day of nuts/walnuts, as part of a low saturated fat and low cholesterol diet and not resulting in increased caloric intake, may reduce the risk of coronary heart disease”. Research in the last two decades has indeed confirmed this beneficial effect for total nuts and walnuts [16], and more recently, a similar qualified health claim was issued for macadamia nuts [112]. Given the shared risk factors between common heart and brain diseases and the manifold salutary effects and safety of nut consumption, and pending additional evidence, this recommendation may also be applied for the prevention of cognitive decline and dementia.

Future large RCTs among healthy pregnant women, children or adolescents, and, particularly, older adults at risk of cognitive decline could provide a new and important public health dimension about simple nutritional recommendations for an entire population. There is a need to develop the area of research on nut consumption and neurodevelopment with a special focus on windows of vulnerability and opportunity, such as the pregnancy, childhood, and adolescence periods, as even if only a small positive cognitive effect is found, this may have significant implications at the population level from a public health perspective. Concerning the evidence currently available, a key methodological limitation is the low accuracy of the screening instruments commonly used to assess cognitive function, such as the MMSE, which compromises the ability to draw firm conclusions. Given that experimental research has tested only bioactives that are present in nuts in isolation, studying the build-up of Aβ plaques and neurofibrillary tangles may provide an avenue to address the limitations of neuropsychological tests. There is a current need to clinically identify if nut interventions may reduce Aβ plaque build-up and tau aggregation by stabilizing microtubules, inhibiting tau phosphorylation, or inhibiting fibrilization. Further research is warranted to elucidate the impact of nut consumption on brain health and better inform cognitive health-related practices and guidelines.

## Figures and Tables

**Figure 1 nutrients-15-01000-f001:**
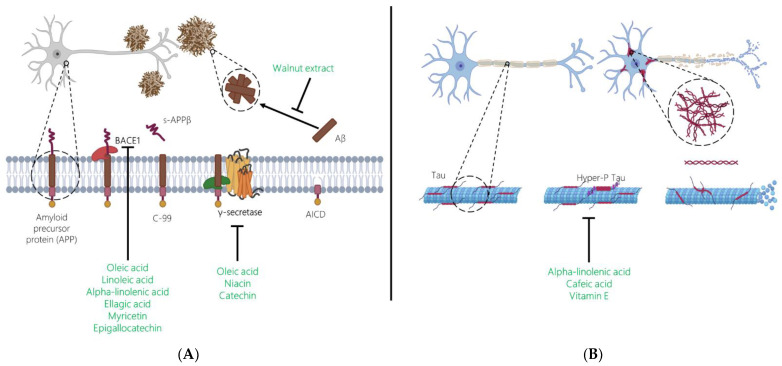
Alzheimer´s Disease Hallmarks (**A**) amyloid-beta (Aβ) deposition and (**B**) neurofibrillary tangles of hyperphosphorylated tau. Effects of nut bioactives on critical pathways are shown.

**Table 1 nutrients-15-01000-t001:** Epidemiological studies examining the association of nut consumption with cognitive performance.

Author, Year	Study Design (Source)	N	Age (Years)	Participant Characteristics	Neuro-Psychological Tests	Nut Dose/Day (Range)	FU (Years)	Outcome
Nurk, 2010 [38]	Cross-sectional (Hordaland Health Study)	2031	70–74	Men & women, general population	Complete battery	No consumption to high consumption	NA	No association
Nooyens, 2011 [39]	Prospective (Doetinchem cohort)	2613	43–70	Men & women, general population	Tests of memory, information processing, cognitive flexibility—sum of test scores (global cognition)	Quintiles of consumption	5	Higher nut consumption associated with cognitive flexibility and global cognition at baseline and trend to delayed cognitive decline at follow-up.
Valls-Pedret, 2012 [40]	Cross-sectional (PREDIMED study)	447	55–80	Men & women at high cardiovascular risk	Comprehensive battery	Total nuts (0–60) Walnuts 1 g (0–30)	NA	Walnuts, but not total nuts, associated with better working memory.
Samieri, 2013 [41]	Prospective (Women’s Health Study)	6174	65+	Women	Comprehensive battery, including TICS	Quintiles of nut consumption within the Mediterranean diet	4	No association of nuts with cognitive changes
O’Brien, 2014 [42]	Prospective (Nurses’ Health Study)	15,467	Mean 74	Women from a selected cohort of nurses	TICS	From never/<1/month to ≥5 servings/week	6	Higher long-term total nut intake associated with better average cognitive status for all cognitive outcomes.
Arab, 2015 [43]	Cross-sectional (NHANES)	5562 and 2975	2 groups: 20–59 ≥60	Men and women, general population	Various cognitive tests	Walnuts with high certainty/walnuts with other nuts	NA	Walnut consumption positively associated with cognitive function in the two groups.
Dong, 2016 [44]	Cross-sectional	894	50 to >80	Men and women from a population cohort	MoCa test	Tertiles of consumption	NA	Higher nut consumption associated with delayed memory. Cognitively healthy adults consumed more nuts than those with MCI.
De Amicis, 2018 [45]	Cross-sectional	279	>65	Men and women attending Nutrition center	MMSE	Highest vs. lowest nut consumption within the Mediterranean diet	NA	OR = 0.30; 95% CI, 0.13–0.69 of low MMSE
Li, 2019 [46]	Prospective	4822	>55	China Health and Nutrition survey	TICS	Consumers of nuts (mainly peanuts)	15	Nuts >10 g/d: OR 0.60, 95% CI 0.43–0.84) of poor cognition
Rabassa, 2020 [47]	Prospective	119	>65	InChianti population study	MMSE	Consumers vs. non-consumers	3	OR: 0.78; 95% CI: 0.61–0.99 of low MMSE
Tan, 2021 [48]	Cross-sectional (NHANES)	1848	60+	Men and women, general population	CERAD total, delayed recall, animal fluency and digit-symbol substitution test	4 groups, from no consumers to consumers meeting recommendations (>30 g/d)	NA	Cognitive scores higher from moderate intake (15.1–30.0 g/d), same in high intake
Jiang, 2021 [49]	Prospective	16,737	Mean 53,5	Singapore Chinese Health Study	MMSE	Nuts <1 serv/mo, 1–3 serv/mo, 1 serv/wk, and =>2 serv/wk	20	3 highest categories: 12% (CI 2–20%), 19% (CI 4–31%) and 21% (CI 2–36%) lower risk of cognitive impairment
Bishop, 2021 [50]	Prospective	3632	65+	Health and Retirement & Health Care and Nutrition studies	TICS	None, low or moderate intake of walnuts	4	Any walnut consumption had greater scores at baseline. No association with cognitive changes.
Chen, 2021 [51]	Cross-sectional	819	70–90	Sydney Memory and Ageing Study:	Comprehensive battery	Consumption of nuts and legumes	NA	Higher consumption related to global cognition (β = 0.117; CI: 0.052–0.181), visuospatial function (β = 0.105; CI: 0.047–0.163), and language (β = 0.113; CI: 0.038–0.189).
Li, 2022 [52]	Prospective	9028	Mean 69	Zhejiang Ageing and Health Cohort Study	MMSE (repeated)	None, <70 g/week, or =>70 g/week	6	Less cognitive impairment (RR = 0.83, 95% CI 0.75–0.91) for highest nut intake group

Abbreviations: CERAD, Consortium to Establish a Registry for Alzheimer’s Disease; CI, confidence interval; FU, follow-up; MCI, mild cognitive impairment; MMSE, Mini-Mental State Examination; mo, month; MoCa, Montreal cognitive assessment (short-term memory recall ability, visuospatial abilities, executive functions, phonemic fluency ability, verbal abstraction ability, attention, concentration and working memory, language, and orientation); N, number of study participants; NA, not applicable; NHANES, National Health and Nutrition Examination Survey; OR, odds ratio; PREDIMED, PREvención con DIeta MEDiterránea; TICS, Telephone Interview for Cognitive Status, a telephone-adaption of the MMSE to assess overall cognitive status; serv, serving.

**Table 2 nutrients-15-01000-t002:** Randomized controlled trials of nuts with outcomes on cognitive performance in adults.

Author, Year	Study Design (Source)	N	Age (Years)	Participant Characteristics	Neuro-Psychological Tests	Nut Dose/Day (Range)	FU	Outcome
Martinez- Lapiscina, 2013 [55]	Parallel (Sub-sample of PREDIMED study)	522	55–80	Men & women at high cardiovascular risk	MMSE and Clock Drawing Test (Administered once at the end of the study)	Mixed nuts, 30 g with MedDiet	6.5 years	MedDiet + nuts: better global cognition compared to control diet.
Valls-Pedret, 2015 [56]	Parallel (Sub-sample of PREDIMED study)	334	55–80	Men & women at high cardiovascular risk	Tests of memory, executive function, global cognition (Administered at baseline and end of study)	Mixed nuts, 30 g with MedDiet	4.1 years	MedDiet + nuts: better memory and a tendency to improved executive function and global cognition compared to control diet.
Barbour, 2017 [57]	Crossover	61	Mean 65	Men and women with overweight/obesity	Tests of memory, executive function, and processing speed	High-oleic acid peanuts, 56–84 g	12 weeks	Short-term memory and verbal fluency improved with the peanut diet compared to control diet.
Dhillon, 2017 [58]	Parallel	86	Mean 31	Men and women with overweight/obesity	Tests of memory and attention	Almonds, dry-roasted at 15% energy	12 weeks	Cognition similarly improved with the almond and control diets.
Sala-Vila, 2020 [59]	Parallel (WAHA study)	708	63–79	Cognitively healthy	Complete test battery	Walnuts at 15% energy	2 years	No effect on cognitive scores in the whole cohort.
Mustra Rakic, 2022 [60]	Parallel	60	50–75	Healthy adults	CANTAB	Almonds/day: 1.5 oz, 3 oz or 3.5 oz snacks	6 months	No among-group changes in cognitive measures.

Abbreviations: CANTAB, Cambridge neuropsychological test automated battery: taps several cognitive domains, including memory, processing speed, and attention; FU, follow-up; MedDiet, Mediterranean diet; MMSE, Mini-Mental State Examination; N, number of study participants; PREDIMED, PREvención con DIeta MEDiterránea. WAHA, Walnuts And Healthy Aging trial.

## Data Availability

Not applicable.

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
