# Peer review of "Impact of Nut Consumption on Cognition across the Lifespan"

_nutrients, 2023, doi:10.3390/nu15041000_

Round 1

Reviewer 1 Report

In the present narrative review the authors discussed about the epidemiological, clinical trial, and mechanistic evidence of the effect of exposure to nuts on cognitive performance. The topic is interesting and of value in the field and the review is well write. Nevertheless, the manuscript need minor  revision before its publication. First of all, to support the effect of the MD on the cognitive Health it would be useful to add recent publication in this field from Baldassano et al. (Fighting the Consequences of the COVID-19 Pandemic: Mindfulness, Exercise, and Nutrition Practices to Reduce Eating Disorders and Promote Sustainability. Sustainability. 2023; 15(3):2120). Furthermore there is some typos like in line 222 (Pribis and cols.);please check carefully all the manuscript. Since you have reported the results of numerous studies that investigated the effects of eating nuts, it might be useful to provide suggestions on how to take them in the final section: quantity, ferquency, differences by type of nuts based on the unsaturated fat acid content.

Author Response

In the present narrative review the authors discussed about the epidemiological, clinical trial, and mechanistic evidence of the effect of exposure to nuts on cognitive performance. The topic is interesting and of value in the field and the review is well write. Nevertheless, the manuscript needs minor revision before its publication.

Thank-you for the feedback and suggestions for improvement. The manuscript has been revised to address the comments provided as summarized below.

The corresponding changes and refinements made are summarized in our response below and in the revised manuscript via track changes.

In the black colour text: the Reviewer’s comments.

In the blue coloured text: response and taken actions.

In the blue coloured text and yellow highlighted notation: significative changes incorporated in the manuscript.

First of all, to support the effect of the MD on the cognitive Health it would be useful to add recent publication in this field from Baldassano et al. (Fighting the Consequences of the COVID-19 Pandemic: Mindfulness, Exercise, and Nutrition Practices to Reduce Eating Disorders and Promote Sustainability. Sustainability. 2023; 15(3):2120).

Thank-you very much for this article suggestion. While it is an interesting read, given this is a review article and the focus is not specific to the Mediterranean Diet, we respectfully suggest not including this reference as there are many references along similar lines or that may be more applicable, which could have been included. The references noted include those based on original findings and a systematic review with the topic being investigated and summarized specific to the Mediterranean diet, cognition, and brain morphology.

Furthermore there is some typos like in line 222 (Pribis and cols.);please check carefully all the manuscript.

Thank-you for this feedback for improvement. The short form has been written in full (highlighted in yellow) as follows:

Page 5, Line 232-233. “To date, the only known RCT specifically investigating the effect of nut consumption on cognitive health in youth was that conducted by Pribis and colleagues in young”

Since you have reported the results of numerous studies that investigated the effects of eating nuts, it might be useful to provide suggestions on how to take them in the final section: quantity, ferquency, differences by type of nuts based on the unsaturated fat acid content.

Thank-you for these recommendations. As much as we would like to be able to confidently provide this information, and while the evidence suggests there is a potential role for nuts in the maintenance of cognitive health and prevention of cognitive decline in individuals across the lifespan, particularly in older adults and those at higher risk, given current limitations to what evidence is available we did not think it would be appropriate to denote specific suggestions on quantity, frequency, etc. at this time without further investigations. However, given the shared risk factors between common heart and brain diseases and the well know beneficial effects of nuts on cardiovascular outcomes, we may suggest consuming daily one and a half servings of nuts, particularly walnuts, for reducing the risk of heart disease, as stated in qualified health claims by the FDA in 2003 (ref. A below) and 2004 (ref. B below), and expect a similar benefit in reducing the risk of cognitive decline and dementia. See lines 613-626 and last three refs (111-113) in the manuscript.

A) Food and Drug Administration (FDA), Qualified Health Claims: Letter of Enforcement Discretion – Nuts and Coronary Heart Disease, Docket No 02P-0505, FDA, Washington, DC, 2003. http://wayback.archive-it.org/7993/20171114183724/https://www.fda.gov/Food/IngredientsPackagingLabeling/LabelingNutrition/ucm072926.htm (accessed on 11 February 2023).

B) Food and Drug Administration (FDA), Qualified Health Claims: Letter of Enforcement Discretion - Walnuts and Coronary Heart Disease (Docket No. 02P-0292), FDA, Washington, DC, 2004. http://wayback.archive-it.org/7993/20171114183725/https://www.fda.gov/Food/IngredientsPackagingLabeling/LabelingNutrition/ucm072910.htm (accessed on 11 February 2023).

C) Food and Drug Administration (FDA), Qualified Health Claims: Letter of Enforcement Discretion – Macadamia Nuts and Reduced Risk of Coronary Heart Disease, FDA, Washington, DC, 2017. https://www.fda.gov/food/food-labeling-nutrition/qualified-health-claims-letters-enforcement-discretion and https://www.fda.gov/media/106201/download (accessed on 11 February 2023).

Specifically, the manuscript has been modified as follows:

Page 18, Lines 613-626. containing diets influence hard clinical outcomes (e.g., dementia or AD) [110]. On the basis of the strength of evidence on nut/walnut consumption and heart disease, the US Food and Drug Administration issued qualified health claims for nuts in 2003 [111] and for walnuts in 2004 [112] stating “Supportive but not conclusive research shows that eating 1.5 ounces per day of nuts/walnuts, as part of a low saturated fat and low cholesterol diet and not resulting in increased caloric intake, may reduce the risk of coronary heart disease”. Research in the last two decades has indeed confirmed this beneficial effect for total nuts and walnuts [16] and more recently a similar qualified health claim was issued for macadamia nuts [113]. Given the shared risk factors between common heart and brain diseases and the manifold salutary effects and safety of nut consumption, and pending additional evidence, this recommendation may also be applied for the prevention of cognitive decline and dementia.

Reviewer 2 Report

I congratulate the authors on the chosen theme, it being of the utmost importance and relevance nowadays, their review bringing forth a well-rounded image regarding the current knowledge on nut consumption and cognition

Author Response

Thank-you for the feedback and kind words.

Reviewer 3 Report

This review discusses the relationship between nut intake and cognitive development at each life stage during pregnancy, lactation, childhood, adulthood, and adulthood.However, some points need to be clarified clearly:

1> "Nuts" should be defined clearly. What type of nuts that are researched? How much effective chemical material contains in each type of nuts can protect people from cognitive decline (like AD)?

2>You should figure out other factors also will effect cognitive test result, especially for test subjects in childhood or old age. Please explain how to avoid the influence of other factors in testing the impact of nut consumption.

Author Response

This review discusses the relationship between nut intake and cognitive development at each life stage during pregnancy, lactation, childhood, adulthood, and adulthood.

We greatly appreciate the time spent, and thoughtful comments provided on our submitted manuscript. We have revised our manuscript were applicable according to the provided comments.

The corresponding changes and refinements made are summarized in our response below and in the revised manuscript via track changes.

In the black colour text: the Reviewer’s comments.

In the blue coloured text: response and taken actions.

In the blue coloured text and yellow highlighted notation: significative changes incorporated in the manuscript.

However, some points need to be clarified clearly:

1> "Nuts" should be defined clearly. What type of nuts that are researched? How much effective chemical material contains in each type of nuts can protect people from cognitive decline (like AD)?

Thank-you for this recommendation. The definition and the types of nuts included in the term “nuts” has been added as follows:

Page 2, Lines 88-89. “Nuts (i.e., tree nuts, including almonds, Brazil nuts, cashews, hazelnuts, macadamias, pecans, pine nuts, pistachios, and walnuts; and peanuts) are an integral part of plant-based diets and have an optimal nutrient profile, particularly abundant in anti-inflammatory and antioxidant molecules, such as unsaturated fatty acids, non-sodium minerals, vitamins, and polyphenols”

The possible components in nuts that may protect against cognitive decline have been described in Section 4 (Page 16). For example, on page 16, lines 533-537; page 17, lines 579 & 584; and in Figure 1.

2>You should figure out other factors also will effect cognitive test result, especially for test subjects in childhood or old age. Please explain how to avoid the influence of other factors in testing the impact of nut consumption.

Thank-you for this comment. Other factors that may also affect the cognitive test results, especially in childhood, include parental social class, parity, type of delivery, breastfeeding, birthweight, maternal IQ, maternal mental health, maternal smoking and alcohol intake during pregnancy, clinical history during pregnancy, overall dietary pattern, and stress events.

In adulthood/older age, these confounding factors include age, sex, educational level, body mass index, cardiovascular risk factors (smoking, hypertension, dyslipidaemia), physical activity, and depression. The dietary pattern is also important, as exemplified by reports of adherence to the Mediterranean diet and improved brain health.

These potential confounders are usually adjusted for in epidemiologic studies, while in randomized controlled trials they should be equilibrated between treatment arms, otherwise the results can also be adjusted for them. In the single publication of a young cohort assessed for nut consumption and cognition (ref. 27), results were not adjusted for the confounding factors that may influence adolescent cognition, and a statement in which all these factors are named has been included after discussion of ref.27 (Page 4, lines 188-193). On the other hand, most reports of adult/older cohorts concerning diet and cognition summarized in the present review include data adjusted for known risk factors influencing cognition. A note in this regard has been added in the corresponding section (Page 6, lines 279-283).

These modifications have been incorporated into the manuscript as follows (indicated by yellow highlights):

Page 4, Lines 188-193. A limitation of this study is that authors did not adjust data for covariables known to influence cognitive performance in youth, such as parental social class, parity, type of delivery, breastfeeding, birthweight, maternal intellectual quotient (IQ), maternal mental health, maternal smoking and alcohol intake during pregnancy, clinical history during pregnancy, overall dietary pattern, and stress events [27].

Page 6, Lines 279-283. The known factors influencing cognitive performance in adulthood, namely age, sex, educational level, body mass index, cardiovascular risk factors (smoking, hypertension, dyslipidemia), physical activity, overall dietary pattern, and depression [3] were treated as confounders and adjusted for in analyses of data from epidemiological studies.

Reviewer 4 Report

Stephanie et al reviewed the nut consumption and their impact on cognition under the title “Impact of Nut Consumption on Cognition Across the Lifespan”. Overall, it is a very well written, balanced narrative review in an unbiased manner. Still, there is no quantitative conclusive evidence for nut consumption in elderly would decrease the risk for dementia. However, as the saying goes ‘food is the better medicine’, it is an important review showing to date literature findings and clinical trials on nut consumption for cognitive improvements.  

I don’t have any major comments but curious to know about in vivo studies for nut consumption.  

Are there no in vivo studies correlating the amelioration of Aβ plaque and/or NFTs by nut consumption? Currently the part of the review contains mostly  in vitro cell lines studies showing benefits for nut derived bioactive molecules. I recommend adding such studies if available.

Are there any comparative quantitative studies (both in vivo and in vitro) deciphered the effect on congnition for different type of nuts. In other words, why walnut is special? What bioactive molecule is enriched in it compared to others?   

Author Response

Stephanie et al reviewed the nut consumption and their impact on cognition under the title “Impact of Nut Consumption on Cognition Across the Lifespan”. Overall, it is a very well written, balanced narrative review in an unbiased manner. Still, there is no quantitative conclusive evidence for nut consumption in elderly would decrease the risk for dementia. However, as the saying goes ‘food is the better medicine’, it is an important review showing to date literature findings and clinical trials on nut consumption for cognitive improvements.  

Thank-you for your review and curiosity. Please find below the details as well as the corresponding refinement to the manuscript made via track changes.

In the black colour text: the Reviewer’s comments.

In the blue coloured text: response and taken actions.

In the blue coloured text and yellow highlighted notation: significative changes incorporated in the manuscript.

I don’t have any major comments but curious to know about in vivo studies for nut consumption.  

Are there no in vivo studies correlating the amelioration of Aβ plaque and/or NFTs by nut consumption? Currently the part of the review contains mostly in vitro cell lines studies showing benefits for nut derived bioactive molecules. I recommend adding such studies if available.

Thank-you for sharing this curiosity. To our knowledge, no one has explored the association between self-reported nut consumption and Alzheimer’s Disease (AD) biomarkers, only adherence to the Mediterranean diet (which includes nuts).

Regarding adherence to the Mediterranean diet and AD, there are both cross-sectional and prospective studies on adherence to Mediterranean diet and burden of AD biomarkers (mostly amyloid) (Vassilaki et al., 2018; Ballarini et al., 2021; Matthews et al., 2014; Berti et al., 2018; Rainey-Smith et al., 2018; full references below). However, none of them specifically dealt with self-reported nut consumption.

Within the manuscript, we have made the following modification:

Page 16, Lines 521-524. “To the best of our knowledge, while there has been exploration of the association between adherence to the MedDiet and AD biomarkers [80], there has been no assessment of nut consumption and AD, and there are no available clinical nut supplementation studies on this specific topic.”

The statement above references the following citation:

Díaz G, Lengele L, Sourdet S, Soriano G, de Souto Barreto P. Nutrients and amyloid β status in the brain: A narrative review. Ageing Res Rev. 2022 Nov;81:101728. doi: 10.1016/j.arr.2022.101728. Epub 2022 Aug 30. PMID: 36049590.

Cross-sectional studies linking adherence to MedDiet with low burden of AD biomarkers

  • Med Diet and amyloid exploring isolated items - BUT not exploring nuts by itself. Vassilaki M, Aakre JA, Syrjanen JA, Mielke MM, Geda YE, Kremers WK, Machulda MM, Alhurani RE, Staubo SC, Knopman DS, Petersen RC, Lowe VJ, Jack CR, Roberts RO. Mediterranean Diet, Its Components, and Amyloid Imaging Biomarkers. J Alzheimers Dis. 2018;64(1):281-290. doi: 10.3233/JAD-171121. PMID: 29889074; PMCID: PMC6031931.
  • MedDiet and markers of AD - BUT not exploring isolated items. Ballarini T, Melo van Lent D, Brunner J, Schröder A, Wolfsgruber S, Altenstein S, Brosseron F, Buerger K, Dechent P, Dobisch L, Duzel E, Ertl-Wagner B, Fliessbach K, Freiesleben SD, Frommann I, Glanz W, Hauser D, Haynes JD, Heneka MT, Janowitz D, Kilimann I, Laske C, Maier F, Metzger CD, Munk M, Perneczky R, Peters O, Priller J, Ramirez A, Rauchmann B, Roy N, Scheffler K, Schneider A, Spottke A, Spruth EJ, Teipel SJ, Vukovich R, Wiltfang J, Jessen F, Wagner M; DELCODE study group. Mediterranean Diet, Alzheimer Disease Biomarkers and Brain Atrophy in Old Age. Neurology. 2021 May 5;96(24):e2920–32. doi: 10.1212/WNL.0000000000012067. Epub ahead of print. PMID: 33952652; PMCID: PMC8253566.
  • Med Diet and amyloid - BUT not exploring isolated items. Matthews DC, Davies M, Murray J, Williams S, Tsui WH, Li Y, Andrews RD, Lukic A, McHugh P, Vallabhajosula S, de Leon MJ, Mosconi L. Physical Activity, Mediterranean Diet and Biomarkers-Assessed Risk of Alzheimer's: A Multi-Modality Brain Imaging Study. Adv J Mol Imaging. 2014 Oct;4(4):43-57. doi: 10.4236/ami.2014.44006. PMID: 25599008; PMCID: PMC4294269.

Prospective studies linking adherence to MedDiet with less deposition of AD biomarkers

  • Med Diet and amyloid - BUT not exploring isolated items. Berti V, Walters M, Sterling J, Quinn CG, Logue M, Andrews R, Matthews DC, Osorio RS, Pupi A, Vallabhajosula S, Isaacson RS, de Leon MJ, Mosconi L. Mediterranean diet and 3-year Alzheimer brain biomarker changes in middle-aged adults. Neurology. 2018 May 15;90(20):e1789-e1798. doi: 10.1212/WNL.0000000000005527. Epub 2018 Apr 13. PMID: 29653991; PMCID: PMC5957301.
  • Med Diet and amyloid, exploring isolated items, but not exploring nuts by itself. Rainey-Smith SR, Gu Y, Gardener SL, Doecke JD, Villemagne VL, Brown BM, Taddei K, Laws SM, Sohrabi HR, Weinborn M, Ames D, Fowler C, Macaulay SL, Maruff P, Masters CL, Salvado O, Rowe CC, Scarmeas N, Martins RN. Mediterranean diet adherence and rate of cerebral Aβ-amyloid accumulation: Data from the Australian Imaging, Biomarkers and Lifestyle Study of Ageing. Transl Psychiatry. 2018 Oct 30;8(1):238. doi: 10.1038/s41398-018-0293-5. PMID: 30375373; PMCID: PMC6207555.

Are there any comparative quantitative studies (both in vivo and in vitro) deciphered the effect on congnition for different type of nuts. In other words, why walnut is special? What bioactive molecule is enriched in it compared to others?

No experimental or clinical studies have been performed to specifically compare two or more different types of nuts on cognitive performance. Walnuts have been studied perhaps more often than other nuts experimentally and clinically, and they are indeed special because of their high content in alpha-linolenic acid (close to 10% by weight, while other nuts only have 1% or less). This was emphasized in the original text (lines 225-231).